# Obesity and Androgen Receptor Signaling: Associations and Potential Crosstalk in Breast Cancer Cells

**DOI:** 10.3390/cancers13092218

**Published:** 2021-05-06

**Authors:** Nelson Rangel, Victoria E. Villegas, Milena Rondón-Lagos

**Affiliations:** 1Departamento de Nutrición y Bioquímica, Facultad de Ciencias, Pontificia Universidad Javeriana, Bogotá 110231, Colombia; 2Centro de Investigaciones en Microbiología y Biotecnología-UR (CIMBIUR), Facultad de Ciencias Naturales, Universidad del Rosario, Bogotá 11001000, Colombia; 3School of Biological Sciences, Universidad Pedagógica y Tecnológica de Colombia, Tunja 150003, Colombia; sandra.rondon01@uptc.edu.co

**Keywords:** obesity, breast cancer, androgens, androgen receptor, adipokines, adiponectin, leptin

## Abstract

**Simple Summary:**

Despite increased information showing obesity is an important breast cancer (BC) risk factor, the mechanisms implicated in this association are not well understood. In this review we describe multiple lines of evidence indicating that altered secretion of androgens and adipokines, from dysfunctional adipose tissue, are independently linked with BC development. However, adipokines (adiponectin and leptin) participate in important biological processes in BC cells by modulating signaling pathways similar to those used by the androgen receptor. These similarities suggest that crosstalk between these factors can occur, with a high probability that its interactions may be responsible for modifying the behavior of normal and tumor cells, especially in obesity. The knowledge of how adiponectin and leptin can interact with the androgen receptor signaling may prospectively guide the development of therapeutic approaches aimed at potentiating the inhibitory actions of adiponectin and androgen receptor or interfering with the pro-stimulatory role of leptin in BC.

**Abstract:**

Obesity is an increasing health challenge and is recognized as a breast cancer risk factor. Although obesity-related breast cancer mechanisms are not fully understood, this association has been linked to impaired hormone secretion by the dysfunctional obese adipose tissue (hyperplasic and hypertrophic adipocytes). Among these hormones, altered production of androgens and adipokines is observed, and both, are independently associated with breast cancer development. In this review, we describe and comment on the relationships reported between these factors and breast cancer, focusing on the biological associations that have helped to unveil the mechanisms by which signaling from androgens and adipokines modifies the behavior of mammary epithelial cells. Furthermore, we discuss the potential crosstalk between the two most abundant adipokines produced by the adipose tissue (adiponectin and leptin) and the androgen receptor, an emerging marker in breast cancer. The identification and understanding of interactions among adipokines and the androgen receptor in cancer cells are necessary to guide the development of new therapeutic approaches in order to prevent and cure obesity and breast cancer.

## 1. Introduction

Breast cancer (BC) is the leading cause of death from cancer in women worldwide [1]. Furthermore, obesity is currently viewed as a major global health challenge and has been confirmed by the International Agency for Research on Cancer as a BC risk factor [2]. Although the mechanisms implicated in this association are not well understood, alterations in the production of sex hormones and changes in secretion of bioactive peptides (adipokines) from adipose tissue have been linked as biological processes usually altered during cancer development [3,4].

In contrast, hormonal dependency of BC has been extensively studied (estrogen receptor—ER and progesterone receptor—PgR) and found to have significant prognostic and predictive value [5]. However, the significance of the androgen receptor (AR), is less well known. Important clinical associations between AR and BC have been reported [6,7], but the biological roles of androgens and the AR in the breast remain unclear, thereby limiting the therapeutic use of AR antagonists in the treatment of BC. Previous studies have suggested that obesity may influence BC risk through androgenic signaling [8,9]; however, evidence of relationship between these factors is inconclusive. Therefore, this review aims to describe the current knowledge on the clinical and biological associations between obesity and the AR signaling in BC. In particularly, we will discuss how the adipocyte-secreted factors (ASFs), adiponectin (ADPN) and leptin (LEP), could potentially interact with the AR signaling pathway to modify the behavior of normal and malignant mammary epithelial cells.

## 2. Obesity and BC

A body mass index (BMI) ≥ 30 Kg/m^2^ is commonly defined as obesity, which is considered an epidemic disease as ~10% of the global population is obese [10], and the projections for some countries indicate that nearly one in two adults will be obese by 2030 [11]. Several studies have demonstrated a link between obesity and a higher risk of health complications, such as type 2 diabetes, heart disease, respiratory problems and cancer. For BC specifically, various systematic reviews and meta-analyses have reported that obese postmenopausal women have an approximately 1.1–1.7-fold increased risk of BC development, whereas an inverse association was observed in premenopausal women [12,13,14,15,16]. This was confirmed in almost 760,000 premenopausal women, showing that an increased BMI is associated with a reduced risk of premenopausal BC [17]. Additionally, obese postmenopausal BC patients have poor outcomes associated with adverse clinical characteristics, such as a larger tumor size or high-grade tumors [18,19,20]. Interestingly, most studies have shown that obesity tends to be correlated with ER negative (ER−) BC, even in premenopausal women [21,22,23]. However, others authors have reported an increased risk only for ER+ postmenopausal BC [19,24,25].

The relationships among overweight/obesity and intrinsic BC molecular subtypes, have also been investigated. An analysis of 1676 postmenopausal BC women, revealed significant associations between being overweight (BMI ≥ 25) and luminal A, B and basal-like intrinsic subtypes, and patients with basal-like BCs were more likely to be highly obese (BMI ≥ 35), compared with underweight/normal patients [26]. Moreover, whole-genome transcriptome analysis performed in triple negative BC (TNBC) patients, revealed specific gene signatures for both premenopausal and postmenopausal women, which enabled the differentiation between normal weight BC patients and overweight/obese patients [27].

### 2.1. Biological Associations

The pathogenesis of BC involves interactions between malignant cells and the mammary microenvironment, and therefore the tumor phenotype may be regulated by not only endocrine but also paracrine signals in the surrounding stromal tissue, which is mainly formed by adipocytes in BC. In this regard, the dysfunctional obese adipose tissue (hyperplasic and hypertrophic adipocytes) secretes high amounts of molecules and extracellular vesicles, which can target cells and promote cell-to-cell communication [28,29]. Elucidating these processes have allowed multiple mechanisms linking obesity with BC initiation and progression to be suggested. Among them, one indicates that hypertrophic adipocytes can trigger extracellular matrix (ECM) remodeling through the secretion of matrix metalloproteinases and increased expression of collagen biogenesis enzymes [30]. The high degree of adipose tissue ECM plasticity also promotes angiogenesis and hypoxic states, which are crucial phenomena related to the release of pro-inflammatory cytokines (e.g NF-κB and HIF-1α), cell growth and tumor survival in BC [31,32] (Figure 1).

The obesity-related hypoxic and inflammatory state is also associated with a decreased response to insulin and insulin-like growth factors (IGFs) signaling, leading to insulin resistance and hyperinsulinemia [33]. In BC cells, higher levels of insulin and IGFs signaling, promote the overstimulation of MAPK and PI3K/AKT pathways [29], both of which are related to increased cell proliferation, survival [34], angiogenesis [35], migration and epithelial to mesenchymal transition (EMT) [36]. Furthermore, the PI3K/AKT signaling regulates glucose metabolism, but cancer cells with impaired insulin function become glucose dependent and switch to relying on aerobic glycolysis for energy production rather than oxidative phosphorylation. This process, known as the Warburg effect, is notably increased in the obese microenvironment and supplies anabolic precursors to fuel the rapid growth and proliferation of BC cells [37]. Interestingly, these obesity-related mechanisms associated to BC, may be explained by dysfunctional secretion of adipokines, which include over 600 hormones and signaling molecules that work in an autocrine, paracrine or endocrine manner [38]. Among them, ADPN, LEP, Visfatin, Resistin, Vaspin, Cytokines (e.g., NFkB, TNFα, IL6) and Progranulin (PGRN) have been related with cancer promotion [39,40]. For example, it has been reported that over-expression of PGNR mediates insulin resistance and obesity. Moreover, it stimulates migration, invasiveness and *VEGF* expression in BC cells. All of this may be explained by the PGRN ability of regulate PI3K/AKT and JAK-STAT signaling [41,42]. In addition, serum PGRN levels were associated with disease progression, therapy response and survival in patients with metastatic BC [43]. Nonetheless, ADPN and LEP will be extensively reviewed below, since they are currently the most important adipokines associated with cancer. In fact, increasing epidemiological and biological studies are investigating the correlation between their altered levels and functions with BC development (Figure 1).

#### 2.1.1. ADPN and BC

ADPN is a monomeric glycoprotein able to trimerize forming a low molecular weight (LMW) isoform, then trimers can combine to form middle (MMW—hexamers) or high molecular weight (HMW until 18-mers) isoforms. Furthermore, ADPN may exist in a globular form, consisting only of the C-terminus domain [44]. ADPN isoforms are found at high concentrations in human serum (5–30 µg/mL) but are reduced in people with a BMI > 25 [45,46]. Lower ADPN concentrations have been established as a risk factor for BC in pre- and postmenopausal women, which was recently confirmed by a meta-analysis of ~30 studies and more than 7000 cases [47] (Figure 1).

In vitro studies have demonstrated the anti-proliferative effects of ADPN on ER+ and ER− BC cells [48,49,50], but the mechanisms by which ADPN isoforms mediates protective roles against BC are not fully defined. ADPN binds to its classical receptors AdipoR1 and AdipoR2 and through recruitment of the adaptor protein APPL1 and the Ser/Thr kinase LKB1, it promotes the activation of the adenosine monophosphate-activated protein kinase (AMPK) [51] (Figure 2). AMPK is an energy sensor, that regulates protein and lipid metabolism by responding to alterations in energy supply and mediates the growth, survival and drug resistance of BC cells [52,53]. AMPK activation induces a strong inhibitory effect on the pro-tumorigenic MAPK and PI3K/AKT pathways [54]. Through AMPK signaling, ADPN further inhibits aromatase activity, which reduces E2 production, ER stimulation and BC cell proliferation [55]. Finally, ADPN may potentially activate PPARα, a nuclear receptor associated not only with fatty acid oxidation, reduced expression of inflammatory genes and suppression of WNT/β-catenin signaling, but also related to BC cell growth and survival [56,57] (Figure 2).

Although ADPN is considered to reduce cellular viability, several studies have shown that ADPN may also increase the proliferation of ER+ cells and even negatively interfere with AMPK activation [58,59,60,61,62]. In fact, treatment of ER+ BC cells with ADPN levels similar to those observed in obese patients triggers a multiprotein complex (AdipoR1/APPL1/IGF-IR/c-SRC) responsible for activating MAPK, which subsequently transactivates the ER to translocate to the nucleus and upregulate estrogen-dependent genes to promote cell growth [58,59,60] (Figure 2). In agreement with these results, pre-clinical studies using transgenic MMTVPyVmT (mouse mammary tumor virus-polyoma middle tumor-antigen) models of cancer, have displayed a pro-angiogenic contribution of ADPN to enhanced mammary tumor growth in vivo [63].

#### 2.1.2. LEP and BC

LEP protein, encoded by the obese (*Ob*) gene, is expressed in several tissues but mainly in breast adipose tissue, possibly explaining why serum LEP levels are higher in women compared with men [64]. Although LEP can participate in a wide range of biological activities, it is primarily involved in regulating body weight by promoting diminuend caloric intake and reducing fat tissue storage through its appetite suppressant actions [65]. Nonetheless, LEP levels are highly increased in obese individuals and LEP is overexpressed in BC patients compared with healthy women [66,67] (Figure 1). In fact, most epidemiological reports and meta-analyses indicate that higher serum LEP levels are significantly associated with an augmented risk of BC development, mainly in postmenopausal patients [68,69]. Furthermore, LEP has been reported as a useful biomarker to differentiate BC patients according to its clinical characteristics (e.g., type, grade and stage) and also to classify tumors depending on their ER status, as ER− cases preferentially have higher LEP levels [70].

In contrast to ADPN, almost all studies have demonstrated that LEP exhibits pro-carcinogenic effects, which are mediated by activation of the obese receptor, a transmembrane protein with at least six alternatively spliced forms (Ob-Ra to Ob-Rf) [71,72,73,74,75]. These neoplastic effects might be explained by the relationship between the overexpression of both LEP and Ob-R and the enhanced activation of classical signaling pathways. Specifically, it has been demonstrated that LEP stimulate BC cell cycle progression and survival by upregulating *MYC*, *CD1*, *BCL2* and *TERT* expression via JAK2/STAT3 activation [73,76]. LEP also promote the expression of the well-recognized oncogenes *JUN* and *FOS* by activating ERK1/2, p38 and JNK, which are typical members of the MAPK signaling pathway [77,78]. In ER+ BC, LEP may stimulates aromatase expression to increase E2 synthesis and further, induce direct functional activation of the ER through STAT3 and ERK-mediated phosphorylation [79]. Moreover, through STAT3 and PI3K/AKT pathways, LEP can upregulate the expression of key markers of EMT, angiogenesis and metastasis (e.g., E-cadherin, Vimentin, *VEGF*, *PKM*2 and *ACAT2*) [30,36,80,81,82,83], in both ER+ and ER− BC cells (Figure 3).

Finally, LEP is considered one of the most important obesity-associated proinflammatory adipokines. It has been reported that higher LEP levels may stimulate NF-κB to increase the expression of pivotal inflammatory mediators such as TNF, IL-1b, and COX-2 [84]. This is associated with enhanced recruitment of infiltrating immune cells (e.g., macrophages) to the mammary microenvironment, thus promoting BC progression [32]. All of the pro-carcinogenic effects attributed to LEP have been confirmed using inhibition assays inhibiting or blocking it (e.g., RNA interference or monoclonal antibodies) which abolish LEP-induced oncogenic effects and BC cell growth in both in vivo and in vitro models [36,85].

## 3. AR Signaling and BC

Approximately 75% of all BCs are ER+. Therefore, most of BC research has focus on identifying the targets and roles of this steroid receptor and its main ligand E2. However, an extensive amount of evidence suggests that estrogens are not the only steroid molecules related to BC [86,87,88], and it has even been shown that hormonally regulated transcription is remarkably maintained in BC subsets with low levels of ER expression or in patients with ER− BC tumors [89,90]. These insights indicate the involvement in BC of additional steroid hormones and its receptors, such as the AR, which is widely expressed in normal breast epithelial cells.

Androgenic hormones are the major circulating sex hormones in both sexes [91]. In females, the ovaries and the adrenal cortex are the main sources of dehydroepiandrosterone (DHEA), dehydroepiandrosterone-sulfate (DHEA-S) and androstenedione (4-dione), which are the major sex steroid precursors of the most potent active metabolites targeting AR (testosterone (T) and 5α-dihydrotestosterone (DHT)) [92]. Several clinical data points to androgens as circulating molecules associated with an increased risk of BC development in pre- and postmenopausal women [86,93,94,95]. However, a recent study revealed that independently of menopausal status, T and DHEA-S serum levels are not useful to predict BC prognosis [96].

In contrast, 80–90% of ER+ tumors are also AR-positive (AR+) and associated with favorable prognoses (e.g., longer relapse-free survival, lower tumor grade, and smaller tumor size) [97,98], which has been confirmed in several meta-analyses [99,100,101]. Regarding ER− BC, up to 31% of them are reported to be AR+ [102] and although some experimental studies and meta-analyses show that AR positivity is associated with improved outcomes [101,103,104], other studies have reported contradictory results [7,105,106]. Despite the above, recent evidence have shown that AR pathway activity is increased in all BC subtypes compared with normal breast tissue [107], making this signaling pathway an interesting target in the study of mammary tumor development and progression.

### 3.1. Biological Associations

AR signaling it is recognized as a master regulator of gene programs associated with a wide variety of biological processes including reproduction, differentiation, cell proliferation, apoptosis, inflammation, metabolism and homeostasis [108]. Androgens and AR employ two main mechanisms in order to exert their functions: (i) dimerization and nuclear translocation of the ligand-stimulated AR that promotes AR–DNA interactions on androgen response elements (AREs) of its target genes [109], and (ii) a non-genomic signaling/action, where androgens can directly activate GPCRs or even regulate the binding of ligand-stimulated AR to cytoplasmic and membrane-bound proteins. Both processes result in the consequent activation of second messengers (e.g., c-SRC) that may induce classical signal transduction cascades [110] (Figure 4).

Different reports have shown that DHEA, DHEA-S, 4-dione and their derivates, can influence the proliferative capacity of breast epithelial cells [86,111]. Furthermore, endogenous aromatization of T (to produce E2) or DHT treatment stimulates the growth of ER+ and ER− BC cell lines [112,113,114]. The same pro-tumorigenic effect was also observed for 4-dione and androstenediol in different BC models [88]. However, some authors reported that androgens, such as 4-dione, DHT and even T, inhibit the proliferation of both ER+ and ER− BC cells [115,116]. Of note, by accurately quantifying steroid hormones, Moon et al. confirmed that the androgens mentioned above are most abundant in BC tumors but show a large variation among patients [117]. This finding, together with the fact that the molecular mechanisms by which androgens control biological processes are not completely known and that the age of menarche and menopause is variable among patients [118], explain the contrasting data reported and more importantly highlight the need for additional basic and functional research to clarify this topic.

#### 3.1.1. The AR in ER+ BC

Many in vitro studies have shown that AR signaling consistently inhibits the basal and estrogen-induced proliferation and survival of MCF-7, T47D and ZR-75-1 cell lines (classical models of luminal BC) [115,116,119]. Its potent anti-proliferative effects were confirmed in MCF-7 cells transfected to overexpress the wild-type or a constitutively active form of the AR [120]. These cells exhibited downregulated expression of *CCND1* and *PTEN* and activation of *CDKN1A*, *TP53* and *BRCA1*, which are well-recognized oncogenes and tumor-suppressor genes, respectively [121,122,123]. At the cellular level, the AR and ER colocalize, providing evidence of signal crosstalk between them [120]. In support of this idea, competition studies for co-regulatory molecules, have demonstrated that in the presence of low AR levels, steroid receptor co-activators (e.g., ARA70) predominantly bind to ER, thereby increasing ER-induced proliferation. Following AR overexpression, co-activators preferentially interact with the AR to antagonize ER signaling [124]. Furthermore, a study using ZR-75-1 cells, displayed reciprocal interference between DHT and E2-induced transcriptional programs, suggesting that AR may antagonize ER signaling by competing with ER for binding to estrogen response elements (EREs) [125] (Figure 4). Inhibitory effects of the AR on ER activity have also been observed through E2 downregulation as AR actions are involved in aromatase transcription decrease [126].

The effects of AR signaling on cell growth however appear to be highly variable in MCF7 cells. Some authors have demonstrated that the AR can maintain a proliferative cell status [110,127,128,129,130] and even suggest that high AR expression may contribute to endocrine-therapy resistance (to both tamoxifen and aromatase inhibitors-AI) [131,132,133,134]. This appears to be explained by activation of the AR and ER non-genomic mechanisms that promote the activation of signaling cascades associated with proliferation and hormone therapy resistance [129,130] (Figure 4). In line with the positive effect of the AR on cell growth, it has been demonstrated that high AR and low ER protein expression levels (AR/ER ratio ≥2) are associated with unfavorable clinical features, tamoxifen resistance and poor survival in ER+ BCs [135,136]. Additionally, ER+ tumors with increased AR expression levels, are characterized by enhanced expression of cell proliferation markers and preferentially classified as luminal B or HER2-enriched tumors [90], which are recognized as more aggressive molecular subtypes.

As AR signaling have shown to inhibits estrogen-induced proliferation and survival of ER+ BC cells, natural and synthetic steroidal androgens were used as therapeutic approach, however they were discontinued for induce serious side effects [137]. Furthermore, the AR agonist enobosarm, a selective AR modulator (SARM), have been proposed to reduce tumor growth in preclinical studies [138], but a clinical trial using different doses of it did not confirm previous results, when administered in postmenopausal ER+ BC patients (NCT02463032). On the other hand, according to the positive effects of AR on ER+ cell proliferation, preclinical studies have confirmed that antiandrogens (i.e., bicalutamide and enzalutamide) inhibit ER+ BC growth by abrogating AR nuclear translocation. However, a clinical trial showed that bicalutamide in combination with AI did not have clinical benefit rates (CBR) in patients with ER+ BC and AI resistance [139]. In contrast, recent results from an additional clinical trial, indicate that in ER+ BCs a subset of women with both high levels of *AR* mRNA and low levels of *ESR1* (ER) mRNA, may benefit from enzalutamide in combination with the AI exemestane [140]. This confirm our previous reports which indicate that high AR levels respect to ER levels may help to identify and treat a specific subgroup of ER+ patients (luminal BC) having tumors with unfavorable clinical and molecular characteristics [90,136].

The above results support the idea that the AR and androgens (DHEA, 4-dione, T, DHT), may play a proliferative role in ER+ tumors. However, combining the data from in vivo and in vitro studies, it appears that a fine-tuning regulation of androgens and estrogens levels, as well as of their receptors, is essential to keep balanced its functions in mammary tissue. Accordingly, two insights could explain the contradictory effects of AR positivity on ER+ BCs: (i) the AR appears to have context-dependent roles as it shifts from an anti-proliferative stimulus to a pro-proliferative one, depending on the high and low expression levels of ER, respectively, and (ii) although AR positivity likely reflects well-differentiated BCs and is thus associated with a good prognosis, the biological effects of AR signaling, when over-stimulated by genomic or non-genomic mechanisms, may confers worse outcomes in BC, which has been recently suggested [141].

#### 3.1.2. AR in ER− BC

In line with the effects of AR in tumors with low ER levels, in vitro studies have shown that the stimulation of the AR signaling can promotes the proliferation of ER− BC cell lines with high (MDA-MB453) or low AR levels (MDA-MB231, MFM-223) [89,142,143]. This was confirmed by siRNA mediated AR knockdown and treatment with anti-androgenic drugs (enzalutamide and bicalutamide), which significantly inhibited the proliferation of MDA-MB453 and other TNBC cells [143,144]. These experiments further showed increased apoptosis and changes in the morphology of ER− cells, which were related to decreased rates of migration and invasion [145] (Figure 4).

Androgen-induced growth of MDA-MB453 cells has been shown to be associated with activation of the WNT/β-catenin and HER2 pathways [135]. These signaling pathways have the ability of stimulate MAPK and PI3K cascades, leading to the upregulation of mitogenic signals from AR. Additionally, it has been reported that MDA-MB453 cells appear to have a high frequency of PI3K mutations [146], and an association between AR, EGFR and ERK1/2 activity has also been identified in TNBC, as treatment with inhibitors of these molecules decreased the amount of AR and had an additive anti-proliferative effect [147]. Besides, in MDA-MB453 cells, the AR can bind to the promoters of genes related to mitogenic effects, that in MCF7 cells are normally occupied by ER [144,148] (Figure 4).

Regarding biological behaviors, extensive molecular profiling and clustering studies have identified great heterogeneity within ER− tumors (basal-like BC subtypes) [149]. Among them, the luminal AR (LAR) subtype, is associated with a transcriptomic profile enriched for AR expression signaling pathways and for having a luminal-like gene profile [89,143,150,151]. LAR tumors are characterized to be highly aggressive [152,153], which has emphasized the need for clarifying the role of AR in BC and promoted the development of several AR antagonists that are currently being tested in multiple clinical trials alone or in combination with other treatment strategies for ER− BC, which have been comprehensively reviewed by other authors [108,154,155,156]. To note, a phase II clinical trial comprising 424 TNBC patients, showed bicalutamide provide good results and have a considerable CBR at 6 months (19%) [157]. Another single-agent phase II clinical trial assessed the effectiveness of enzalutamide in 75 patients with AR+ TNBC, showing that CBR was 28% at 24 weeks [158]. Particularly, LAR subtype cells frequently carry PI3K mutations [146,159]. Thus, in a phase IB/II clinical trial, Lehmann et al., demonstrated that the combination of enzalutamide with taselisib, a PI3K inhibitor, significantly increased the CBR of TNBC patients (35.7%), but among them, LAR subtype patients trended towards better response compared to non-LAR (75.0% vs. 12.5%, *p* = 0.06) [160]. In general, few side effects have been reported from all these strategies, indicating its potential use for women with TNBC.

## 4. Obesity, AR Signaling and BC

As mentioned above, the ovaries and the adrenal cortex are the main sources of androgens, but in postmenopausal women these sex steroids are mainly synthesized by adipose tissue [161,162]. This suggests that obesity may contribute to a hyperandrogenic state, and as the breast is largely constituted by adipose tissue, the increased local androgen production should not be ignored in BC research. In fact, a pioneering study reported higher DHEA concentrations in the mammary adipose tissue of obese subjects, even in the adjacent adipose tissue of malignant BC compared with benign tumors [163]. Moreover, overweight BC patients with high T levels, clearly have worse prognoses [164], and a combined harmful effect of high serum T concentrations and a non-healthy dietary pattern on BC risk has been reported [165]. In agreement with this, the Nurses’ Health Study showed that weight gain and higher BMIs (>25) are associated with an increased risk of developing AR+ tumors in postmenopausal women [104].

Biologically, the increased androgen synthesis induced by obesity may favors the growth of ER+ cells due to T aromatization and augmented E2 production. However, it must also be assumed that the proliferative stimulus may be enhanced by the itself elevation of androgens and their interaction with AR. In line with this insight, a recent study demonstrated that in a preclinical model of obesity, AR transcriptional activity was increased and promoted ER+ BC progression in environments with low E2 availability (i.e., in postmenopausal cases). Thus, in models treated with enzalutamide (anti-androgenic drug), AR inhibition prevented obesity-associated tumor progression by suppressing the proliferation and survival of BC cells [166]. Hence, high AR expression in obesity could play a positive role in BC progression when ER expression levels are low. Consistent with this idea, patients with basal-like BCs (ER− or low expression) are more likely to be obese, compared with underweight/normal patients [26], and tumors with high AR and low ER expression levels (AR/ER ratio ≥2) are associated with more aggressive BC molecular subtypes [90,136]. Together these results allow suggest that in obesity ER expression may decrease, opening the possibility that the increased androgen synthesis observed in this environment, results in specific overstimulation of AR signaling to maintain active pathways related to tumor progression and aggressiveness.

### 4.1. Adipokines and the AR: Potential Links in BC

Although it has been reported that postmenopausal obese women have elevated circulating or mammary adipose tissue androgens levels compared with lean women, some studies have found inconsistent results [166]. Therefore, AR signaling modulation cannot be entirely explained by the discrepancy in androgens levels between obese and lean individuals. Among other factors regulating AR signaling, the adipokine IL-6 has been shown to sensitize ER+ and ER− BC cells to T, as the expression of the *AR* and its target gene *FKBP5*, were significantly increased. Bioinformatic analyses further demonstrated that *AR* and *FKBP5* expression were higher in ER+ BC cases with increased IL-6 levels [166]. To the best of our knowledge, direct associations between the AR signaling pathway and important adipokines, such as ADPN and LEP, have not been yet reported in BC cells, but similarly to IL-6, their action mechanisms lead us to believe that functional interaction among them and AR, cannot be ruled out.

#### 4.1.1. ADPN and AR

Most studies indicate that AR signaling plays a positive role in ER+ BC prognoses by blocking ER function. In this context, ADPN has been reported to prevent androgens aromatization [55], which involves increased levels of androgenic hormones in mammary tissue, AR overstimulation and increased inhibition of the ER signaling. Therefore, the decreased levels of ADPN, observed in obesity and cancer, may favor ER-driven BC progression by reducing the activation of the AR signaling. In ER− tumors, in which AR activation is associated with sustained growth, low ADPN levels may decrease the inhibition of the AR signaling, thereby maintaining the proliferative status of this BC subtype (Figure 5).

ADPN can activate multiple intracellular signaling pathways. Through AMPK-independent signaling, ADPN promotes interactions with TKRs, such as the IR, which may increase PI3K and MAPK signaling in both ER+ and ER− BCs [59,60]. These signaling pathways may be enhanced by crosstalk between TKRs and the membrane-bound AR form, which has been reported to activate these classical pathways [58,61,167]. Furthermore, several ligand-independent phosphorylation sites in the AR protein have been identified to act as regulatory hot-spots capable of sensing cellular signaling and guiding biological responses in BC cells [129,130]. In fact, c-SRC, AKT, ERK1/2 and other MAPK-pathway associated factors, reported to be regulated by ADPN [168], can promote the ligand-independent activation of AR via phosphorylation at its N-terminal and hinge region domains [167]. All of these AR-dependent non-genomic mechanisms may serve as potential links to ADPN in BC (Figure 5).

It cannot be excluded that ADPN functions in BC cells, might also be performed through the direct regulation of *AR* gene expression. In fact, several pioneer and general transcription factors (e.g., AP-1, CREB, FOXA1 and SP1) target the *AR* promoter and can be regulated by pathways stimulated by ADPN [169]. It has been reported that some adipokines, can suppress androgen sensitivity and the proliferation of androgen-dependent LNCaP cells (prostate cancer) through *AR* downregulation [170]. A similar behavior in BC, by decreased levels of ADPN in obesity, would support the inhibitory effect of ADPN in ER− or low ER expression postmenopausal cases, in which AR signaling plays a clearer role as a tumor promoter. In this cellular environment, opposite effects on *AR* expression might induce BC cell growth, consistent with the reported tumorigenic actions of ADPN [58,59,60]. In an ER+ BC context, the downregulation of *AR* expression by decreased levels of ADPN would promote tumor growth because the AR would not be enabled to inhibit ER signaling (Figure 5).

Accordingly, ADPN has the potential to transactivate the *AR*, which has been reported for the ER [59,60]. However, it must be considered that the specific effects of ADPN effects on AR signaling may vary depending on the changes in ADPN levels, as observed in both obesity and BC.

#### 4.1.2. LEP and AR

Although it has been reported that androgens can modulate LEP expression in adipocytes [171], the effects of LEP on AR signaling in normal and malignant mammary epithelial cells have not been described. However, several studies have confirmed that increased levels of LEP and AR are found in obese-postmenopausal BC cases [32,68,70,163,166] and both have been identified as oncogenic factors that use similar signaling pathways. Therefore, a functional crosstalk among them cannot be excluded.

Upon LEP binding, the Ob-Rs trigger downstream cascades in two different ways, (i) via phosphorylation of JAK/STATs or (ii) by activation of several adaptors or membrane-associated proteins (e.g., GRB2, SOS, SHP2 and c-SRC) [172]; Both mechanisms are independently identified as elements that targets and promote ligand-independent activation of the membrane-bound AR form [167,173,174] (Figure 6). In support of this insight, previous data have shown that STAT3 activation enhances its interaction with the AR N-terminal domain in prostate cancer cells stimulated with the adipokine IL-6 [175,176,177,178]. In ER− or low ER expression BC cells, if a similar JAK/STAT-mediated interaction between the AR and LEP occurs, presuppose an activation of the AR pro-tumorigenic signaling (non-genomic effects), as well as, its nuclear translocation to further enhance cell growth (genomic effects). Conversely, considering the anti-tumorigenic effect of the AR in ER+ BCs, these interactions would involve inhibition of AR signaling to promote the oncogenic roles of the ER, furthering agreeing with the ability of LEP to induce increased aromatase expression [78,79,179]. Additionally, the AR has been shown to phosphorylate HER2, and interactions between Ob-R and HER2 have also been reported, both leading to the subsequent activation of JAK/STATs, PI3K/AKT and MAPK signaling [81,180]. This crosstalk could be specifically mediated by the membrane-bound AR form as all of these factors can activate adaptor proteins like c-SRC. Therefore, the activation of non-genomic AR mechanisms by either Ob-R or HER2 would promote the oncogenic signaling of each other (Figure 6).

Of note, STAT-family members are transcription factors itself that bind to a gamma-activated sequence or an interferon-stimulated response element, and according to the Signaling Pathways Project web knowledge base, these elements have been identified in the *AR* gene promoter region [181,182]. This suggests that the LEP-mediated activation of STAT family members could directly regulate *AR* gene expression, similar to IL-6 [166]. Indeed, a recent study reported that *AR* expression was downregulated in LNCaP cells cultured in the presence of high LEP concentration [183]. In luminal BC cells, the observation of a similar effect on *AR* gene expression, has led to the hypothesis that high LEP levels might potentiates the oncogenic effects of the ER by reducing AR levels and its inhibitory action on the ER. In contrast, opposite effects on *AR* expression could be observed in ER− or low ER expression cells, which would be consistent with the confirmed pro-tumorigenic role of both, LEP and the AR, in basal-like BC subtypes [74,103] (Figure 6).

## 5. Concluding Remarks and Future Perspectives

BC is a heterogenous disease, and its development and progression depends on several risk factors. Of these, obesity is an increasing health challenge that certainly modifies the behavior of normal mammary tissue by dysfunctional secretion of molecules from hyperplasic and hypertrophic adipocytes. Therefore, in addition to endocrine signals originating from distal adipose tissues, the direct effects of paracrine signals from the local adipose tissue in breast microenvironment should be a primary research focus. Among the ASFs, altered ADPN and LEP levels have been shown to be independently associated with BC development. LEP functions are clearly pro-tumorigenic, whereas ADPN actions appear to be ER-status dependent. In this regard, it will be challenging to determine how the opposing actions of ADPN promote growth in ER+ cells but inhibit ER− cells. Other molecules produced by adipose tissue include androgens, which are steroid hormones with altered levels in obese and BC cases. Similar to ADPN, AR signaling has been shown to have ER context-dependent roles. Specifically, AR activation promotes ER− BC cell proliferation, but inhibition of the E2/ER-stimulated growth. Regardless of the specific end-effect of ADPT, LEP, or androgen/AR on BC cells, it has been clearly reported that their actions are mainly modulated by similar intracellular signal transduction pathways. These similarities suggest that crosstalk between adipokines and AR signaling can occur, and that there is a high probability that these interactions are responsible for modifying the behavior of normal and malignant mammary epithelial cells, especially in obesity.

To better understand how modified levels of ADPT, LEP, and AR influences BC development and progression, improved in vitro (e.g., co-culture experiments) and in vivo (e.g., tissues or cell lines xenograft specimens) experimental obesity models are required to study their interactions. These approaches will allow the evaluation of not only the degree to which the adipokines and the AR promote activation or inactivation of each other, but also whether their coordinated actions can modulate the biological processes that they are involved in. Because the epidemiological data are inconsistent, regarding the association of obesity with hormone receptor expression, menopausal status and BC subtypes, it will be interesting to clarify the relationship between lifestyles and clinical associations. To address this topic, further epidemiological studies with a larger number of included patients are needed. Additionally, the assessment of potential correlations among adipokines with androgen levels or AR status will be useful to establish their influence on BC prognosis.

Clearly, obesity is a biological variable that merits consideration regarding the therapeutic strategies used to manage BC. Accordingly, and because of the modifiable nature of obesity, the most recommendable approach must be to promote lifestyle changes (in pre- and postmenopausal women) to achieve a healthy weight to prevent cancer development and even improve BC treatment efficacy. In line with our literature review, it is logical to assume that drugs targeting ADPN, LEP and even adipokines less studied as PGRN, require a deeper evaluation on their use in BC management. However, regarding AR, clinical trial results indicate that antiandrogen therapies may be effective and needed for AR+ TNBC patients, especially for LAR subtypes or even non-LAR with AR-weak positivity. Furthermore, although AR inhibition seems not to be widely effective in ER+ patients, focus need to be put in the identification of luminal tumors where AR levels are higher respect to ER levels. Preclinical and clinical research suggest that these tumors, as well as AR+ HER2-Enriched BC subtypes, are more dependent on androgen signaling. Therefore, additional studies using antiandrogen therapies, alone or in combination with other strategies, should be considered as an important medicinal target, that in the future, will allow to obtain significant clinical benefits for patients with different BC subtypes.

The knowledge of how ADPN and LEP can interact with the AR signaling pathway in breast epithelial cells may prospectively guide the development of new therapeutic approaches aimed at potentiating the inhibitory actions of the antiandrogen therapies in BC. Nevertheless, the use of drugs targeting ADPN and the AR as potential therapeutic tools, will have to be carefully assessed and separately studied according to the ER status, because of the contradictory effects reported of they in each BC tumor subtype.

## Figures and Tables

**Figure 1 cancers-13-02218-f001:**
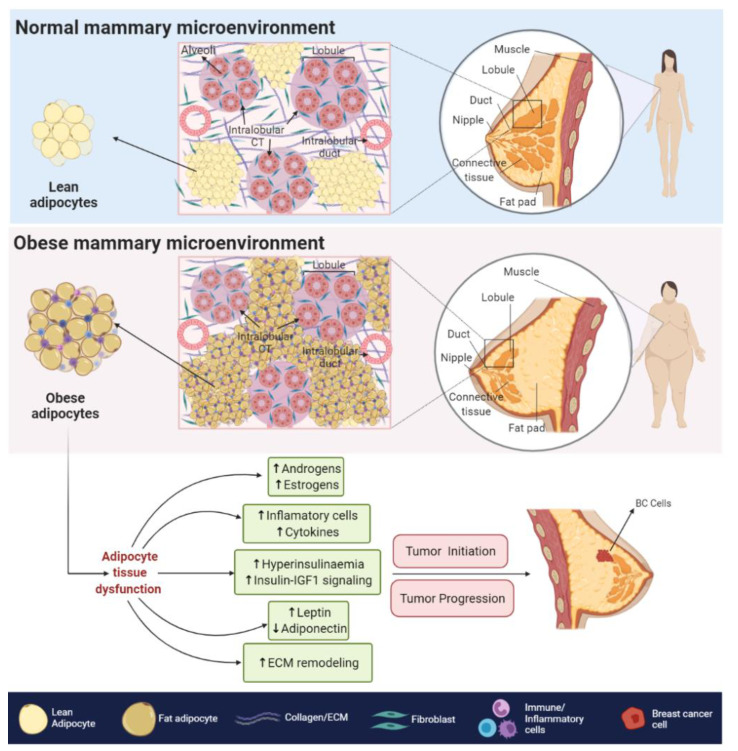
Normal vs. obese mammary microenvironment. Altered biological processes and production of ASFs are observed in obesity and are associated with BC development. CT—connective tissue; ECM—extracellular matrix.

**Figure 2 cancers-13-02218-f002:**
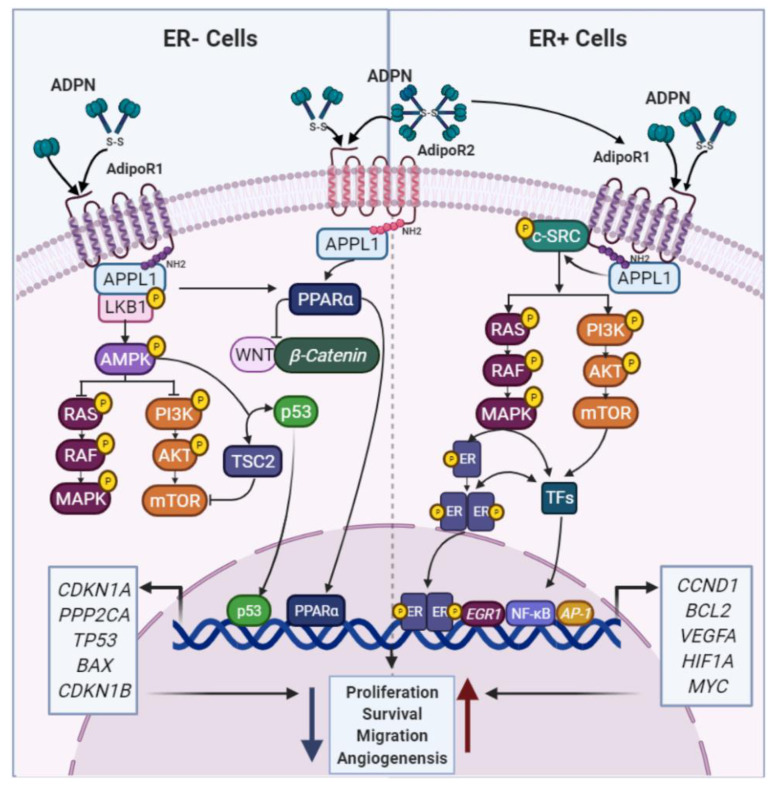
ADPN effects in BC cells. ADPN isoforms and their receptors regulate several signaling pathways to control BC cell behavior. However, it has been reported that ADPN can have anti- or pro-oncogenic effects depending on the ER status. PPARα—peroxisome proliferator activated receptor alpha; TSC2—tuberous sclerosis 2; TFs—Transcription factors; EGR1—early growth response 1; *CCND1*—cyclin D1; *BCL2* apoptosis regulator; *VEGF*—vascular endothelial growth factor; *HIF1A*—hypoxia inducible factor 1 subunit alpha; *CDKN1A*—cyclin dependent kinase inhibitor 1A; *PPP2CA*—protein phosphatase 2 catalytic subunit alpha; *TP53*—tumor protein p53; *BAX*—BCL2 associated X, apoptosis regulator; *CDKN1B*—cyclin dependent kinase inhibitor 1B.

**Figure 3 cancers-13-02218-f003:**
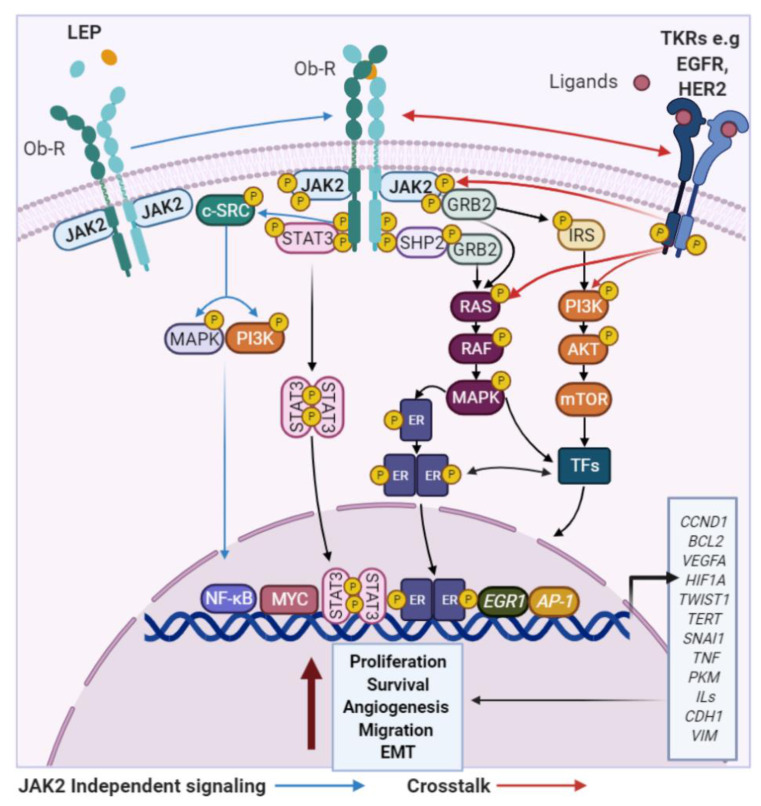
LEP effects in BC cells. LEP and Ob-Rs activate classical signaling pathways to regulate the expression of genes related to increased BC development independently of the ER status. TKRs—tyrosine kinase receptors; EGFR—epidermal growth factor receptor; JAK2—janus kinase 2; STAT3—signal transducer and activator of transcription 3; GRB2—growth factor receptor bound protein 2; SHP2—protein tyrosine phosphatase non-receptor type 11; IRS2—insulin receptor substrate 2; *TWISTS1*—twist family bHLH transcription factor 1; *TERT*—telomerase reverse transcriptase; *SNAI1*—snail family transcriptional repressor 1; *TNF*—tumor necrosis factor; *PKM*—pyruvate kinase M1/2; *ILs*—Interleukins; *CDH1*—E-cadherin; *VIM*—vimentin.

**Figure 4 cancers-13-02218-f004:**
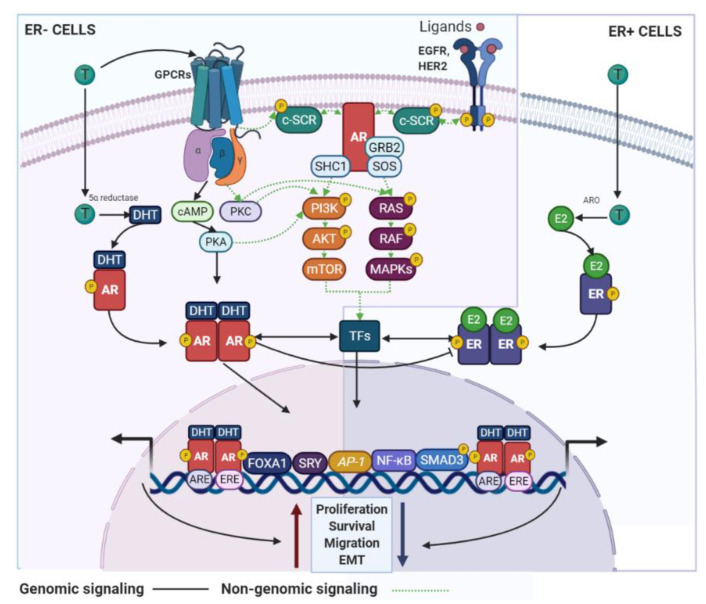
AR signaling in BC cells. AR use genomic and non-genomic mechanisms that can regulate BC development. These signaling pathways are associated with inhibitory or stimulatory effects on the BC cell growth and are dependent of the ER status. GPCR—G protein-coupled receptors; SHC1—SHC adaptor protein 1; SOS—SOS Ras/Rac guanine nucleotide exchange factor; ARO—aromatase.

**Figure 5 cancers-13-02218-f005:**
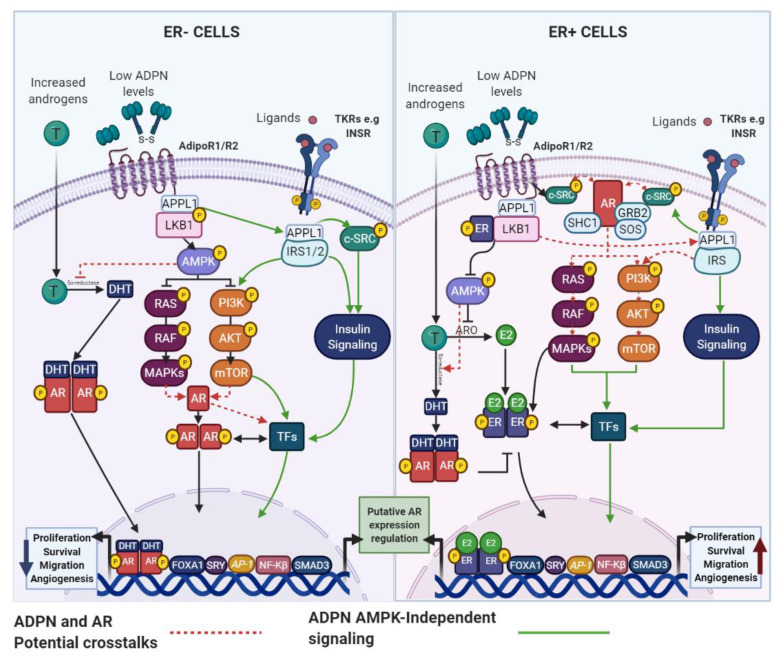
Potential links between ADPN and AR signaling. ADPN and the AR activate common signaling pathways and the crosstalk between them may be involved in regulating BC development, that is dependent on the ER status. INSR—insulin receptor; SRY—sex determining region Y.

**Figure 6 cancers-13-02218-f006:**
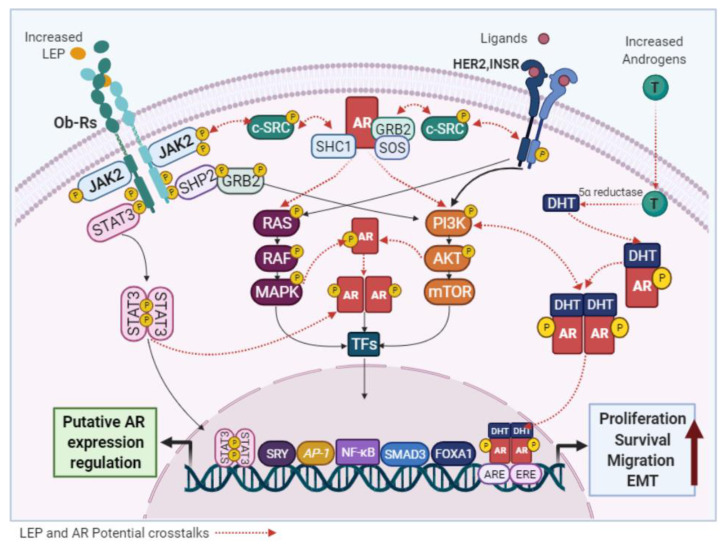
Potential links between LEP and AR signaling. LEP and the AR activate common signaling pathways and the crosstalk between them, may be responsible for the pro-stimulatory effects of LEP observed in BC cells.

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
