# Peer review of "Obesity and Androgen Receptor Signaling: Associations and Potential Crosstalk in Breast Cancer Cells"

_cancers, 2021, doi:10.3390/cancers13092218_

Round 1

Reviewer 1 Report

This is an interesting and timely Review focusing on an under-represented topic in breast cancer research.  The figures are informative and well done.  Moderate grammatical editing is required.  The individual subsections cover a comprehensive review of the field on obesity and androgen receptor signaling.  The addition of more clinical and translational discussion would give this Review a stronger impact and citation value.

Author Response

Bogotá, March 18 - 2021

Dear 
Prof. Dr. Samuel C. Mok 
Editor-in-Chief
Cancers

Please find enclosed the revised version of the review article entitled "Obesity and androgen receptor signaling: associations and potential crosstalk in breast cancer cells”, by Rangel and co-workers.

We have accepted all suggestions, as specified by a point-by-point response, as requested. All amendments have been highlighted in blue color in the manuscript file. We acknowledge that the revision following the raised criticisms highly improved the quality of the manuscript.

We thank you for your attention and hope that this work may now fulfil the scientific standards of Cancers

Sincerely yours,

Nelson Rangel
Corresponding author

REVIEWER(S)' COMMENTS TO AUTHOR

REVIEWER 1

This is an interesting and timely Review focusing on an under-represented topic in breast cancer research.  The figures are informative and well done.  Moderate grammatical editing is required.  The individual subsections cover a comprehensive review of the field on obesity and androgen receptor signaling.  The addition of more clinical and translational discussion would give this Review a stronger impact and citation value.
AUTHORS: We thank the reviewer for finding our manuscript interesting and timely. Regarding moderate grammatical editing, we confirmed that the English of our manuscript was edited by the Edanz group (www.edanzediting.com/ac). If the reviewer considers that this latest version of our manuscript requires a new edition, we will gladly do it with the Edanz group, for which we require 7 business days to carry out such an edition. We are looking forward to your comments.
    In addition, following the reviewer's suggestions related to adding more clinical and translational discussion, we have increased the discussion on the results of clinical trials conducted to evaluate antiandrogen therapy in different subtypes of breast cancer. This information was included on page 8 (lines 305-321) and page 9 (lines 345-351 and lines 359-370).

Reviewer 2 Report

The review is written comprehensively. It may be published.

Author Response

Bogotá, March 18 - 2021

Dear 
Prof. Dr. Samuel C. Mok 
Editor-in-Chief
Cancers

Please find enclosed the revised version of the review article entitled "Obesity and androgen receptor signaling: associations and potential crosstalk in breast cancer cells”, by Rangel and co-workers.

We have accepted all suggestions, as specified by a point-by-point response, as requested. All amendments have been highlighted in blue color in the manuscript file. We acknowledge that the revision following the raised criticisms highly improved the quality of the manuscript.

We thank you for your attention and hope that this work may now fulfil the scientific standards of Cancers

Sincerely yours,

Nelson Rangel
Corresponding author

REVIEWER(S)' COMMENTS TO AUTHOR

REVIEWER 2

The review is written comprehensively. It may be published.
AUTHORS: We thank the reviewer for finding our manuscript written comprehensively, and for recommending its publication.

Reviewer 3 Report

This is a well referenced comprehensive review on the influence of altered adipokines production in obesity ( increased leptin and decreased adiponectin ) on androgen receptor signaling in breast cancer.The topic is of interest as obesity is a major risk factor for breast cancer and the androgen pathway holds promise as a therapeutic target even though its rol;e remains to be defined.

The Authors should be congratulated for summarizing a lot of controversial information with numerous references which will be very useful for the readership of the Journal interested in this topic.

The main concern that I have with this review is that it lacks a translational / clinical focus.There are numerous trials of antiandrogen therapy in different subtypes of breast cancer ( even though they may not relate specxifically to obese patients ) which are not discussed.Discussion of the rationale of these trials and overall results should be included in session 3 ( pags 6-9 ) where the authors review AR signaling and Breast Cancer.How do the results of the clinical trials inform us about the translationally relevant signaling pathways in the different subtypes of breast cancer ?

In its present form the review provides a lot of detailed frequently conflicting data ( no fault of the Authors ) but fails to provide specifics of the Authors' view of the big picture,ie how should we move forward to improve the use of antiandrogen therapy based on the data reported  here and the results of the clinical  trials

Author Response

Bogotá, March 18 - 2021

Dear 
Prof. Dr. Samuel C. Mok 
Editor-in-Chief
Cancers

Please find enclosed the revised version of the review article entitled "Obesity and androgen receptor signaling: associations and potential crosstalk in breast cancer cells”, by Rangel and co-workers.

We have accepted all suggestions, as specified by a point-by-point response, as requested. All amendments have been highlighted in blue color in the manuscript file. We acknowledge that the revision following the raised criticisms highly improved the quality of the manuscript.

We thank you for your attention and hope that this work may now fulfil the scientific standards of Cancers

Sincerely yours,

Nelson Rangel
Corresponding author

REVIEWER(S)' COMMENTS TO AUTHOR

REVIEWER 3

1. This is a well referenced comprehensive review on the influence of altered adipokines production in obesity (increased leptin and decreased adiponectin) on androgen receptor signaling in breast cancer. The topic is of interest as obesity is a major risk factor for breast cancer and the androgen pathway holds promise as a therapeutic target even though its role remains to be defined. The Authors should be congratulated for summarizing a lot of controversial information with numerous references which will be very useful for the readership of the Journal interested in this topic.
AUTHORS: We thank the reviewer for finding our manuscript well referenced, interesting and comprehensive.

2. The main concern that I have with this review is that it lacks a translational / clinical focus. There are numerous trials of antiandrogen therapy in different subtypes of breast cancer (even though they may not relate specifically to obese patients) which are not discussed. Discussion of the rationale of these trials and overall results should be included in session 3 (pags 6-9) where the authors review AR signaling and Breast Cancer. How do the results of the clinical trials inform us about the translationally relevant signaling pathways in the different subtypes of breast cancer?
AUTHORS: We thank the reviewer for these important suggestions. Following the reviewer's suggestions, we have increased the discussion on the results of clinical trials conducted to evaluate antiandrogen therapy in different subtypes of breast cancer. This information was included in the page 8 (lines in 305-321) and page 9 (lines 345-351, and lines 359-370).

3. In its present form the review provides a lot of detailed frequently conflicting data (no fault of the Authors) but fails to provide specifics of the Authors' view of the big picture, ie how should we move forward to improve the use of antiandrogen therapy based on the data reported here and the results of the clinical trials
AUTHORS: We thank the reviewer for this important observation. Following the reviewer's suggestion, we have provided our point of view around the information reported in the literature (clinical trials) on the use of antiandrogen therapy in breast cancer patients (page 13, line 537; page 14, lines 538-554).

Reviewer 4 Report

Obesity is an increasing health challenge and is recognized as a breast cancer risk factor. Although obesity-related breast cancer mechanisms are not fully understood, this association has been linked to impaired hormone secretion by the dysfunctional obese adipose tissue. Among these hormones, altered production of androgens and adipokines is observed, and both, are independently associated with breast cancer development.

This review covers the relationships between these factors and breast cancer, focusing on the biological mechanisms by which signaling from androgens and adipokines modifies the behavior of mammary epithelial cells.

In general, the review is well done and covers the literature appropriately. I have some minor suggestions to improve the paper:

  • I would add an additional small section to mention additional adipokine, which may have a role in breast cancer. For example progranulin has been recently identified as a key adipokine Mediating High Fat Diet-Induced Insulin Resistance and Obesity through IL-6 in Adipose Tissue (Cell metabolism, Volume 15, Issue 1, 4 January 2012, Pages 38-50). Because progranulin has a well-established role in breast cancer, it is reasonable to make a connection between progranulin, obesity and breast cancer.
  • Is there any difference in ER cross-talk with adipokine between ER alfa or ER beta expressing cells?
  • Please use italic for gene names and for in vitro and in vivo.

Author Response

Bogotá, March 18 - 2021

Dear 
Prof. Dr. Samuel C. Mok 
Editor-in-Chief
Cancers

Please find enclosed the revised version of the review article entitled "Obesity and androgen receptor signaling: associations and potential crosstalk in breast cancer cells”, by Rangel and co-workers.

We have accepted all suggestions, as specified by a point-by-point response, as requested. All amendments have been highlighted in blue color in the manuscript file. We acknowledge that the revision following the raised criticisms highly improved the quality of the manuscript.

We thank you for your attention and hope that this work may now fulfil the scientific standards of Cancers

Sincerely yours,

Nelson Rangel
Corresponding author

REVIEWER(S)' COMMENTS TO AUTHOR

REVIEWER 4

1. Obesity is an increasing health challenge and is recognized as a breast cancer risk factor. Although obesity-related breast cancer mechanisms are not fully understood, this association has been linked to impaired hormone secretion by the dysfunctional obese adipose tissue. Among these hormones, altered production of androgens and adipokines is observed, and both, are independently associated with breast cancer development.
This review covers the relationships between these factors and breast cancer, focusing on the biological mechanisms by which signaling from androgens and adipokines modifies the behavior of mammary epithelial cells. In general, the review is well done and covers the literature appropriately. 
AUTHORS: We thank the reviewer for finding our manuscript well done.

2. I have some minor suggestions to improve the paper: I would add an additional small section to mention additional adipokine, which may have a role in breast cancer. For example, progranulin has been recently identified as a key adipokine Mediating High Fat Diet-Induced Insulin Resistance and Obesity through IL-6 in Adipose Tissue (Cell metabolism, Volume 15, Issue 1, 4 January 2012, Pages 38-50). Because progranulin has a well-established role in breast cancer, it is reasonable to make a connection between progranulin, obesity and breast cancer.
AUTHORS: We thank the reviewer for this important suggestion. Following the reviewer ‘suggestion, we have added information regarding the role of Progranulin in breast cancer (page 3, lines 113-123).

3. Is there any difference in ER cross-talk with adipokine between ER alfa or ER beta expressing cells?
AUTHORS:  Despite the growing molecular evidence of a functional crosstalk between the leptin and/or adiponectin with ERα, few studies have focused on the interaction between the leptin and/or adiponectin with ERβ at cell growth level in breast cancer cells [1-3].  In fact, some studies by examined the protein levels of estrogen receptors ERα and ERβ, found that ERα protein expression was significantly increased in MCF7 cells treated for 7 days with 100 ng/mL of leptin, whereas ERβ protein levels were decreased. Thus, leptin induces a robust increase in the ERα to ERβ ratio in MCF-7 breast cancer cells [4].
     In agreement, in vivo experiments using a nude mouse xenograft model showed that injection of leptin in the tumor for 15 days, led to upregulation of ERα expression and downregulation of ERβ expression in human breast tumor [5]. Consistently, positive correlation between the expression of leptin receptor and ERα has been reported in human breast cancer biopsies [1], although other authors have not observed such association [6]. In contrast, the association between ERβ and leptin in breast cancer has rarely been reported, with a study showing a positive correlation between leptin/leptin receptor and ERβ expression [6]. All these findings give supportive evidence of the existence of a crosstalk between leptin and ERα, but not between leptin and ERβ. 
    In addition, the association between ERβ and adiponectin in breast cancer has rarely been reported, with a study showing that in ER-positive HC11 cells, recombinant adiponectin increased ERβ expression, inhibited cell proliferation and induced apoptosis. Further, in this study were defined novel synergistic roles for adiponectin in promoting ERα/ERβ cross talk in malignant mammary epithelial cells (MEC) [7]. An additional studie have indicated that adiponectin induces the expression of both ER in MEC [8], suggesting that adiponectin may exert its antitumor effects by regulating the direction of ERα and/or ERβ signaling. However, whether adiponectin similarly functions in normal (nontumorigenic) MEC to influence ERα/ERβ cross talk have not been determined. 

4. Please use italic for gene names and for in vitro and in vivo
AUTHORS: The names of the genes were revised and written in italics, as well as the words in vitro and in vivo were written in italics. 

REFERENCES
1. Fusco R, Galgani M, Procaccini C, Franco R, Pirozzi G, Fucci L, Laccetti P, Matarese G. Cellular and molecular crosstalk between leptin receptor and estrogen receptor-{alpha} in breast cancer: Molecular basis for a novel therapeutic setting. Endocr Relat Cancer 2010; 17: 373-382. 

2. Garofalo C, Sisci D, Surmacz E. Leptin interferes with the effects of the antiestrogen ICI 182,780 in MCF-7 breast cancer cells. Clin Cancer Res 2004; 10: 6466-6475. 

3. Dieudonne MN, Machinal-Quelin F, Serazin-Leroy V, Leneveu MC, Pecquery R, Giudicelli Y. Leptin mediates a proliferative response in human MCF7 breast cancer cells. Biochem Biophys Res Commun 2002; 293: 622-628. 

4. Adamo Valle, Jorge Sastre-Serra, Jordi Oliver and Pilar Roca. Chronic Leptin Treatment Sensitizes MCF-7 Breast Cancer Cells to Estrogen. Cell Physiol Biochem 2011; 28: 823-832. 

5. Yu W, Gu JC, Liu JZ, Wang SH, Wang Y, Zhang ZT, Ma XM, Song MM. Regulation of estrogen receptors alpha and beta in human breast carcinoma by exogenous leptin in nude mouse xenograft model. Chin Med J (Engl) 2010; 123: 337-343. 

6. Garofalo C, Koda M, Cascio S, Sulkowska M, Kanczuga-Koda L, Golaszewska J, Russo A, Sulkowski S, Surmacz E. Increased expression of leptin and the leptin receptor as a marker of breast cancer progression: Possible role of obesity-related stimuli. Clin Cancer Res 2006; 12: 1447-1453. 

7. Omar M. Rahal and Rosalia C. M. Simmen. Paracrine-Acting Adiponectin Promotes Mammary Epithelial Differentiation and Synergizes with Genistein to Enhance Transcriptional Response to Estrogen Receptor β Signaling. Endocrinology 2011; 152: 3409–3421. 

8. Treeck O, Lattrich C, Juhasz-Boess I, Buchholz S, Pfeiler G, Ortmann O. Adiponectin differentially affects gene expression in human mammary epithelial and breast cancer cells. Br J Cancer 2008; 99: 1246 –1250.

Round 2

Reviewer 3 Report

The authors have improved the quality of the Review emphasizing more the clinical aspect of Androgen receptor targeted therapy.